# Decision Regret and Vaccine Hesitancy among Nursing Students and Registered Nurses in Italy: Insights from Structural Equation Modeling

**DOI:** 10.3390/vaccines12091054

**Published:** 2024-09-14

**Authors:** Alice Silvia Brera, Cristina Arrigoni, Silvia Belloni, Gianluca Conte, Arianna Magon, Marco Alfredo Arcidiacono, Malgorzata Pasek, Galyna Shabat, Luigi Bonavina, Rosario Caruso

**Affiliations:** 1Department of Biomedicine and Prevention, University of Rome Tor Vergata, 00133 Rome, Italy; alicesilvia.brera01@universitadipavia.it; 2Department of Public Health, Experimental and Forensic Medicine, University of Pavia, 27100 Pavia, Italy; cristina.arrigoni@unipv.it; 3Health Professions Research and Development Unit, IRCCS Policlinico San Donato, 20097 San Donato Milanese, Italy; gianluca.conte@grupposandonato.it (G.C.); arianna.magon@grupposandonato.it (A.M.); 4Medical Department, University Hospital of Parma, 43126 Parma, Italy; marcoalfredo.arcidiacono@unipr.it; 5Department of Nursing, Faculty of Health, University of Applied Sciences in Tarnów, 33-100 Tarnów, Poland; malgorzata_pasek@wp.pl; 6Division of General and Foregut Surgery, IRCCS Policlinico San Donato, 20097 San Donato Milanese, Italy; galyna.shabat@gmail.com (G.S.); luigi.bonavina@unimi.it (L.B.); 7Department of Biomedical Sciences for Health, University of Milan, 20122 Milan, Italy; 8Clinical Research Service, IRCCS Policlinico San Donato, 20097 San Donato Milanese, Italy

**Keywords:** vaccine hesitancy, decision regret, nursing students, registered nurses, COVID-19 vaccination

## Abstract

This study focused on vaccine hesitancy and decision regret about the COVID-19 vaccine among nursing students (BScN and MScN) and Registered Nurses (RNs) in Italy. The primary aim was to describe decision regret and vaccine hesitancy among these groups and to understand what influences vaccine hesitancy. Data were collected through an e-survey conducted from March to June 2024. The Decision Regret Scale and the Adult Vaccine Hesitancy Scale were employed to assess regret and hesitancy levels, assessing trust, concerns, and compliance regarding vaccination. Among the participants, 8.64% were not vaccinated. The results indicated moderate to high levels of decision regret and diverse levels of trust, concerns, and compliance with COVID-19 vaccination. Structural equation modeling revealed that decision regret significantly predicted Trust (R^2^ = 31.3%) and Concerns (R^2^ = 26.9%), with lower regret associated with higher trust and lower concerns about vaccine safety. The number of COVID-19 vaccine boosters was a significant predictor of Trust and Concerns, with more boosters associated with higher trust and lower concerns. MScN students exhibited higher Compliance compared to RNs (R^2^ = 2.9%), highlighting the role of advanced education. These findings suggest that addressing decision regret and providing comprehensive vaccine information could enhance trust and compliance.

## 1. Introduction

Vaccine hesitancy has emerged as a critical global health issue, significantly impacting the efficacy of immunization programs worldwide [1,2]. Despite the proven effectiveness of vaccines in preventing infectious diseases, a growing number of individuals are reluctant or refuse to be vaccinated due to a variety of factors [3]. Misinformation about vaccine safety and efficacy, often amplified by social media, plays a substantial role in fueling doubts [4]. Additionally, distrust in healthcare systems and authorities, coupled with socio-cultural influences and personal beliefs, further exacerbate vaccine hesitancy [5].

The COVID-19 pandemic has underscored the urgency of addressing vaccine hesitancy [1,3,4]. With the rapid development and deployment of COVID-19 vaccines, concerns about their safety and efficacy have become particularly pronounced [6,7]. Vaccine acceptance varies significantly across different countries, influenced by public trust in government and healthcare institutions, perceived risks and benefits of vaccination, and exposure to misinformation [8]. The global impact of the COVID-19 pandemic makes understanding and mitigating vaccine hesitancy a priority public health strategy to ensure the effectiveness of vaccination campaigns. Globally, vaccine acceptance is 63% in the general population, higher among healthcare workers (64%) and individuals with chronic diseases (69%), but lower among pregnant women (48%) and parents consenting for their children (61.29%), with a pooled vaccine hesitancy rate of 32% [4].

Vaccine hesitancy among students, especially those in healthcare fields, is influenced by a complex interplay of factors that include general misconceptions about vaccines and specific concerns related to their professional roles and personal beliefs. Research indicates that students may harbor doubts about vaccine safety due to misinformation and perceived low risk of disease, leading to reduced vaccine uptake even among those training to be healthcare professionals [9,10,11,12]. Cultural and societal influences, as well as the accessibility of accurate information, play a crucial role in shaping students’ attitudes toward vaccines, not only concerning COVID-19 but also other vaccines [13]. Addressing these factors through targeted educational strategies and integrating vaccine education into the core curriculum of healthcare programs could significantly reduce vaccine hesitancy among future healthcare workers [12,13,14].

Italy has faced significant challenges with vaccine uptake, exacerbated by political debates and fluctuating policies that have further influenced public perception [15,16,17]. Vaccination rates across Italy vary, with disparities evident among different regions [18]. The northern regions generally exhibit higher vaccination rates, whereas the central and southern regions face more pronounced hesitancy and lower uptake [18,19,20]. These discrepancies could be attributed to varying levels of healthcare infrastructure, socio-economic factors, and public trust in health authorities [21].

A systematic review and meta-analysis of 17 studies estimated the COVID-19 vaccine hesitancy rate among healthcare workers in Italy at 13.1% (95% CI: 6.9–20.9%) [22]. Before and during the vaccination campaign, the hesitancy rates were 18.2% (95% CI: 12.8–24.2%) and 8.9% (95% CI: 3.4–16.6%), respectively [22]. Major reasons for vaccine hesitancy included a lack of information, concerns about vaccine safety, and a fear of adverse events. In general, the vaccine hesitancy among healthcare workers in Italy is much lower than the broader European study, which found a pooled COVID-19 vaccine acceptance rate of 66% (95% CI: 61–71%), with significant geographic variability [23]. Focusing particularly on Bachelor of Science in Nursing (BScN) students, Master of Science in Nursing (MScN) students, and Registered Nurses (RNs) is crucial due to their unique position within the healthcare sector [24]. These professionals and future professionals are not only recipients of vaccines but also key influencers of public perception and vaccination behaviors. As frontline health professionals, their attitudes and practices could significantly impact the broader community’s vaccine uptake.

Despite the general understanding of vaccine hesitancy, there is a paucity of research examining decision regret and vaccine hesitancy among nursing students and healthcare workers in Italy [5,24]. Decision regret refers to the distress or remorse individuals feel after making a healthcare decision [25], in this case, receiving a COVID-19 vaccine. This regret could significantly impact future health behaviors and attitudes towards vaccination. There is a notable lack of comprehensive models that integrate determinants such as demographic factors, number of boosters, decision regret regarding vaccination, and regional differences in predicting vaccine hesitancy [26]. This gap highlights the need for targeted research to inform effective interventions for these specific groups, as their role is critical in shaping public perceptions and vaccination behaviors [27]. For this reason, the primary aim of this study is to describe the levels of decision regret and vaccine hesitancy among BScN students, MScN students, and RNs in Italy. As a secondary aim, this study seeks to understand how determinants—such as age, gender, location, and number of boosters—influence vaccine hesitancy. Based on previous evidence, we hypothesize that factors such as the number of boosters, age, gender, and regional location directly and indirectly affect vaccine hesitancy [27]. In this regard, decision regret could play a crucial role in these indirect effects, as feelings of regret may alter trust in healthcare systems, amplify concerns about vaccine safety, and reduce compliance with vaccination recommendations.

## 2. Materials and Methods

### 2.1. Design and Data Sources

This observational study collected data using an electronic survey (e-survey). The reporting of this study was guided by the Consensus-Based Checklist for Reporting of Survey Studies (CROSS), which provides a standardized framework for survey reporting to enhance the reliability and transparency of the research findings [28]. This study was approved by the Institutional Review Board of the University of Pavia (protocol 3/CDS/2024). Data collection was initiated in March 2024 and concluded in June 2024. This project employed a comprehensive survey methodology distributed to a broad demographic of BScN and MScN students, as well as RNs, via the SurveyMonkey online platform. Healthcare organizations and universities were invited to disseminate the survey to reach a wide audience.

### 2.2. Subjects and Sample Size

The sample size for this study was determined using a precision-based approach, taking into account the finite population of healthcare workers and students [29]. Given the population size, including roughly 440,000 RNs and 50,000 students [30], this approach ensures that the sample size is both statistically significant and practical. Specifically, the sample size calculation incorporated the finite population correction to adjust for the relatively large sample size in proportion to the population. The formula used for this calculation was as follows:(1)$$n_{\mathrm{adj}}\;=\;\frac{\frac{Z^{2}\;\cdot \;p\;\cdot \;\left(1\;-\;p\right)}{E^{2}}}{1\;+\;\left(\frac{\frac{Z^{2}\;\cdot \;p\;\cdot \;\left(1\;-\;p\right)}{E^{2}}\;-\;1}{N}\right)}$$

In this approach [29], *n* is the sample size calculated using the proportion of vaccine hesitancy, N is the total population size (roughly 490,000 individuals), *Z* is the *Z*-value (1.96 for a 95% confidence level), *p* is the estimated proportion of vaccine hesitancy (0.14 according to Barello et al. [24] or 0.182 considering the meta-analysis of Renzi et al. [23]), and E is the margin of error (0.05). Using this formula, the required sample size ranges between 227 and 239 individuals, depending on the proportion of vaccine hesitancy used in the calculation.

### 2.3. Survey Development and Content Validation

The survey collected various demographics and professional data, including sex (female, male, other), age (years, median, IQR), marital status (unmarried/living alone, married), and location (Northern Italy, Central Italy, Southern Italy). Participants were also asked about their professional/academic profile, which was categorized as BScN student, MScN student, or worker as RN. For the workers as RNs, work settings were detailed as emergency or critical care, medical care, surgical care, primary care, or outpatient services. Additionally, the survey included questions about COVID-19 vaccination status, asking whether participants had received the vaccine and, if so, the number of boosters (no boosters, one booster, two boosters, three boosters, more than three boosters). The survey incorporated the Decision Regret Scale (DRS) and the Adult Vaccine Hesitancy Scale (aVHS), both of which underwent a validation process [31].

The DRS is a widely used tool for measuring regret associated with healthcare decisions [32]. It consists of five items that assess the degree of regret an individual feels after making a healthcare-related decision. Respondents rate each item on a five-point Likert scale, ranging from 1 (strongly agree) to 5 (strongly disagree). The DRS is designed to capture a single underlying construct of decision regret, making it a unidimensional scale. The DRS was not initially available in Italian and was translated following a structured process, following authorization from the author who developed the scale [32]. This step included initial translation, synthesis of translations, review by a committee, and content validity assessment using the Content Validity Ratio (CVR), as detailed in Appendix A. The employed translation approach followed an established multi-step methodology to gain cultural equivalence in the translation and adaptation [33]. A panel of 10 experts evaluated the relevance and clarity of each item in the Italian context, resulting in a validated Italian version (see Appendix A). This version was used for the survey, and its psychometric validity was confirmed through Confirmatory Factor Analysis (CFA). The CFA demonstrated well-explained sample statistics when the unidimensionality was tested [χ^2^_(5, N=324)_ = 10.636, *p* = 0.0591; RMSEA = 0.059, 90%CI (0.000–0.109), *p* = 0.322; CFI = 0.980; TLI = 0.960; SRMR = 0.034] and a Cronbach’s alpha of 0.896, indicating high reliability. In this study, the DRS scores were obtained by calculating the mean of the item responses after reversing the negatively worded items (item 2 and item 4). This approach ensures that higher DRS scores indicate lower decision regret.

Although the aVHS was previously translated and tested for reliability in Italian [34], its dimensionality was not assessed. Therefore, an exploratory factor analysis (EFA) was conducted using the maximum likelihood robust (MLR) estimator and varimax rotation, as detailed in Appendix A. The number of factors to be extracted was determined by parallel analysis, interpretation of the scree plot, and previous evidence of dimensionality [31]. The three-factor solution was found to be the most plausible, explaining the sample statistics well: χ^2^_(18, N=324)_ = 20.504, *p* = 0.305; RMSEA = 0.021, 90%CI (0.000–0.056), *p* = 0.906; SRMR = 0.012. The three-factor solution was chosen as it provided the best fit for the data and aligned well with theoretical expectations and previous findings. The factors were interpreted as follows: (a) Factor 1: trust in vaccine safety and benefits; (b) Factor 2: concerns about vaccine safety; (c) Factor 3: trust in health authorities and compliance (i.e., compliance). Each factor’s internal consistency was evaluated using McDonald’s ω, given the multidimensional nature of the scale. The internal consistency for each factor was as follows: Factor 1 = 0.87, Factor 2 = 0.82, Factor 3 = 0.80. For the scoring procedure, the mean of the items for each factor was calculated to create a composite score for that factor after reversing items 5, 9, and 10 (see Appendix A). This approach allows for a detailed analysis of different aspects of vaccine hesitancy, providing a nuanced understanding of the respondents’ attitudes. Higher mean scores indicate greater trust (Factor 1) or compliance levels (Factor 3), respectively, while higher scores for Factor 2 indicate lower concerns about vaccine safety.

### 2.4. Survey Administration

In designing our population-based survey, we anticipated achieving a response rate of approximately 50%, a benchmark consistent with the current literature on survey research methodologies [35]. To meet this objective and achieve our sample size target, we strategically targeted approximately 239 respondents by inviting 460 potentially eligible participants from the national territory encompassing Northern, Central, and Southern Italy. The contacts for these participants were obtained through collaborations with healthcare organizations and universities across Italy, selected based on their geographical distribution to ensure a representative sample.

Specifically, we involved three universities and three healthcare organizations from each of Italy’s three major geographic areas: northern, central, and southern regions. We targeted institutions with nursing programs offering BScN and MScN degrees for the university sample. This study’s principal investigator contacted program coordinators at these universities to obtain permission to distribute the questionnaire to their students. These coordinators either provided access to student email lists or directly distributed the questionnaire link to students through their internal communication channels. The universities selected from each region were chosen to reflect the diversity of the student population across different parts of Italy. Similarly, for the sample of RNs, we collaborated with three healthcare organizations from each geographic area, including hospitals, primary care centers, and outpatient services. These organizations were contacted through their human resources or professional development departments to request permission to include their nursing staff in this study. Once permission was granted, the organizations either provided the contact information for eligible RNs or distributed the survey link internally to ensure privacy and adherence to organizational policies.

Prior to the launch of the survey, a comprehensive preparation process was undertaken. This included training team members on the ethical considerations of conducting survey research, familiarization with the survey tool for troubleshooting participant queries, and strategies for effective communication to encourage participation. A pilot test was conducted with 5 respondents to identify and rectify any issues in the survey design or administration process.

The survey was administered through an online platform, SurveyMonkey. This allowed the survey to be accessible 24/7, providing flexibility for participants to engage at their convenience. The recruitment period started in March 2024 and concluded in June 2024. Advanced features of the SurveyMonkey platform were utilized to mitigate the risk of multiple participation in the web-based survey. These features included the use of cookies and IP tracking to prevent individuals from completing the survey more than once. Additionally, the survey was designed to include validation checks for consistency in responses and to flag potential duplicate entries for manual review.

Once a potential respondent received the invitation, an informative sheet was available for them to read before proceeding. Before answering the survey, they were required to electronically sign an informed consent form, ensuring that they understood this study’s purpose, role, and rights as participants [36].

### 2.5. Eligibility Criteria

For BScN and MScN students, participants must be enrolled in a BScN or MScN program in Italy. Additionally, they must have completed at least one semester of their respective program to ensure familiarity with healthcare practices and vaccination information and must be 18 or older. For RNs, participants must be currently practicing as nurses in Italy. They must have at least one year of professional experience in the healthcare sector to ensure substantial exposure to vaccination campaigns and healthcare decision-making. Exclusion criteria for this study include individuals who have participated in a similar survey or study on vaccine hesitancy within the past six months to avoid bias due to prior exposure. Individuals who are unable to provide informed consent, including those with cognitive impairments or language barriers that prevent them from understanding the survey content, are also excluded.

Additionally, participants who have been on extended leave or sabbatical from their nursing programs or professional practice are excluded, as they may not have current or relevant experience with vaccination practices. Individuals who have recently relocated to Italy and have not been engaged in the Italian healthcare system for at least one year are excluded to ensure familiarity with local vaccination policies and practices. Lastly, respondents who fail to complete the survey in its entirety are excluded, as incomplete data may not provide a comprehensive understanding of vaccine hesitancy and decision regret. Even though we targeted potential eligible respondents through our invitations, participants were required to self-assess their eligibility based on the criteria provided upon receiving the invitation.

### 2.6. Data Analysis

Descriptive statistics were performed, and summaries are presented based on the nature and distribution of each variable. The distributions of the aVHS scores and DRS were visualized using violin plots, which included box plots inside the violins and jittering to allow readers to ascertain the distribution of these measures, responding to this study’s main aim. The bivariate relationships between DRS and the three scores of the aVHS were assessed to determine the feasibility of the modeling aimed at testing the hypothesis that sustains the secondary aim. Structural equation modeling (SEM) was employed to test the hypothesized relationships between the potential determinants and vaccine hesitancy. The dependent variables in the model were Trust, Concerns, and Compliance (the scores of the aVHS). The model was specified to include age, location (North Italy, Central Italy, South Italy), profile (BScN students, MScN students, workers as RN), number of boosters (no boosters, one booster, two boosters, three boosters, more than three boosters), sex, and DRS as independent variables. Additionally, the indirect effects of DRS on Trust, Concerns, and Compliance were evaluated to test its ability to mediate the relationships from predictors to scores related to vaccine hesitancy. This approach allowed us to understand both the direct impact of the determinants on vaccine hesitancy and the potential mediating role of decision regret. The SEM analysis provided fit indices to evaluate the adequacy of the model, including χ^2^, root mean square error of approximation (RMSEA), comparative fit index (CFI), Tucker–Lewis index (TLI), and standardized root mean square residual (SRMR). These indices were used to assess the model fit, with criteria indicating a good fit being non-significant χ^2^, RMSEA < 0.06, CFI > 0.90, TLI > 0.90, and SRMR < 0.08. The analysis was performed using R 4.2.2 (R Core Team, 2023) and involved libraries such as Lavaan for SEM, tidyverse, and ggplot2 for visualization. Significance was set at 5%, and two-tailed tests were used for inferential analysis.

## 3. Results

### 3.1. Sample Characteristics

The overall response rate for this study was 71%. As shown in Table 1, the study sample consisted of 324 participants. Of these, 240 (74.1%) were females, 82 (25.3%) were males, and 2 (0.6%) identified as other. The median age of the participants was 22 years, with an interquartile range (IQR) of 20 to 25 years. Regarding marital status, 296 (91.4%) were unmarried or living alone, while 28 (8.64%) were married.

Participants were distributed across various regions of Italy, with 169 (52.2%) from Northern Italy, 68 (21.0%) from Central Italy, and 87 (26.9%) from Southern Italy. The professional profile of the participants included 191 (59.0%) BScN students, 72 (22.2%) MScN students, and 61 (18.8%) RNs.

Among the 61 working RNs, the distribution of work settings was as follows: 7 (11.5%) in emergency or critical care, 13 (21.3%) in medical care, 14 (23.0%) in surgical care, 14 (23.0%) in primary care, and 13 (21.3%) in outpatient services.

Regarding COVID-19 vaccination status, 296 (91.4%) participants reported having received the vaccine. The number of COVID-19 vaccine boosters among the vaccinated participants was distributed as follows: 27 (8.33%) had received one booster, 111 (34.3%) had received two boosters, 139 (42.9%) had received three boosters, and 19 (5.89%) had received more than three boosters. Notably, 28 (8.64%) participants were not vaccinated, including 24 BScN students, 3 MScN students, and 1 RN who cited health reasons for not being vaccinated.

The DRS scores had a median of 2, with an IQR of 1.2 to 2.8. The aVHS scores showed the following distributions: the median score for Trust in Vaccine Efficacy and Benefits was 1.5 (IQR: 1.0–2.0), for Concerns about Vaccine Safety, it was 2.67 (IQR: 2.0–3.33), and for Trust in Health Authorities and Compliance, it was 2.0 (IQR: 1.3–2.3).

### 3.2. Distribution of DRS and aVHS

Figure 1 illustrates the distributions of the DRS and the aVHS scores, including Trust, Concerns, and Compliance.

The Trust scores also display a concentrated distribution, with a majority of the scores falling between 1 and 2. This indicates that most participants have a relatively high level of trust in vaccine efficacy and benefits. The Concerns scores exhibit a wider distribution, with a noticeable spread across the entire range of the scale. The scores are fairly evenly distributed, with a slight concentration around the median, suggesting diverse levels of concern about vaccine safety among the participants. The distribution of the Compliance scores is relatively symmetric, with a concentration of scores around the median value. Most of the scores fall between 1 and 3, with some outliers extending towards higher values, indicating varying levels of compliance among participants.

The DRS scores show a relatively narrow distribution, with most scores clustered around the lower end of the scale (higher regret area). This suggests that, on average, participants experienced moderate–high levels of regret regarding their vaccination decisions.

### 3.3. Bivariate Relationships and Structural Equation Modeling

Figure 2 presents this study’s bivariate relationships among key variables, revealing several significant correlations.

Age showed a significant positive correlation with being male (r_pb_ = 0.117, *p* = 0.035), and older participants were more likely to have received more boosters (r = 0.256, *p* < 0.001). The DRS score was significantly correlated with several variables: Trust (r = 0.521, *p* < 0.001), indicating that lower regret levels (as high scores are lower regret) were associated with higher trust in vaccine efficacy and benefits; Concerns (r = 0.468, *p* < 0.001), suggesting that lower regret was also associated with lower concerns about vaccine safety; Sex (r_pb_ = 0.114, *p* = 0.041), showing a slight positive correlation where females had marginally higher regret levels; and the number of boosters (r = −0.153, *p* = 0.006), where fewer boosters were associated with higher regret scores.

The model fit the sample statistics well, with χ^2^_(10, N=324)_ = 16.373, *p* = 0.089; RMSEA = 0.044, 90%CI (0.000–0.080), *p* = 0.554; CFI = 0.969; TLI = 0.855; and SRMR = 0.030. The results of the SEM analysis are presented in Table 2.

The model explained 31.3% of the variance in Trust. The Decision Regret Score was a significant positive predictor of Trust (Estimate = 0.396, SE = 0.044, *p* < 0.001), indicating that lower regret levels (as high scores are lower regret) were associated with higher trust in vaccine efficacy and benefits. The number of COVID-19 vaccine boosters also significantly predicted Trust, with participants who received one booster (Estimate = −0.531, SE = 0.205, *p* = 0.009), two boosters (Estimate = −0.475, SE = 0.197, *p* = 0.016), three boosters (Estimate = −0.397, SE = 0.197, *p* = 0.045), and more than three boosters (Estimate = −0.483, SE = 0.222, *p* = 0.030) showing lower trust compared to those with no boosters.

The model explained 26.9% of the variance in Concerns. Lower regret (higher scores) was associated with lower concerns (higher scores in factor 2) (Estimate = 0.463, SE = 0.054, *p* < 0.001). The number of COVID-19 vaccine boosters significantly predicted Concerns, with participants who received one booster (Estimate = −0.656, SE = 0.227, *p* = 0.004), two boosters (Estimate = −0.458, SE = 0.189, *p* = 0.015), three boosters (Estimate = −0.575, SE = 0.187, *p* = 0.002), and more than three boosters (Estimate = −0.715, SE = 0.260, *p* = 0.006) showing lower concerns compared to those with no boosters.

The model explained 2.9% of the variance in Compliance. Only being an MScN versus being a worker as an RN was a predictor of Compliance (Estimate = −0.296, SE = 0.155, *p* = 0.050).

The model explained 4.4% of the variance in the DRS score. Receiving more than three boosters (Estimate = −0.527, SE = 0.181, *p* = 0.004), being a BScN student (Estimate = −0.536, SE = 0.176, *p* = 0.002), and being an MScN student (Estimate = −0.366, SE = 0.170, *p* = 0.031) were associated with lower decision regret compared to their respective reference groups.

The indirect effects of the DRS score on Trust, Concerns, and Compliance were not significant (indirect effect on Trust Estimate = −0.048, SE = 0.135, *p* = 0.723; indirect effect on Concerns Estimate = −0.355, SE = 0.235, *p* = 0.130; indirect effect on Compliance Estimate = 0.000, SE = 0.005, *p* = 0.933).

## 4. Discussion

This study highlighted the high levels of decision regret and concerns about vaccine safety among nursing students (BScN and MScN) and RNs in Italy, which were the primary focus of our investigation. The median DRS score of 2.0 on a 5.0 scale indicated moderate to high levels of regret regarding vaccination decisions, while the aVHS scores reflected diverse levels of Trust, Concerns, and Compliance among the participants, and surprisingly, 8.6% of the sample was not vaccinated (one RN, 24 BScN students, and 3 MScN students). These results should be considered when designing educational initiatives for students, RNs, and even the general population.

The SEM analysis provided further insights into the variables influencing vaccine hesitancy. The indirect effects of the DRS score on Trust, Concerns, and Compliance were not significant. This implies that changes in the independent variables (such as the number of boosters) do not affect the dependent variables (Trust, Concerns, Compliance) through the DRS; therefore, DRS does not function as a mediator in this model. Additionally, the model explained 31.3% of the variance in Trust. The DRS score was a significant positive predictor of Trust, indicating that lower regret levels were associated with higher trust in vaccine efficacy and benefits. As expected, the number of COVID-19 vaccine boosters also significantly predicted Trust, with participants who received one booster showing higher trust compared to those with no boosters. These findings align with the previous literature indicating that personal experiences with vaccination can significantly influence trust in vaccines, although the specific impact of decision regret has been less explored in prior studies [37]. However, a research study found that COVID-19 booster vaccine intentions were uncorrelated with the number and intensity of side effects or occurrence of an intense side effect from the primary COVID-19 vaccine [38]. Conversely, trust in the quality and safety of vaccines, trust in vaccine development, and low concern over vaccine side effects are critical for the willingness to be vaccinated [38,39]. Therefore, building a strong sense of trust in vaccination by promoting campaigns aiming to increase booster intentions seems to be the most relevant factor [38].

The path from predictors to Concerns aligned with the anticipated results: lower regret (higher DRS score) was associated with lower concerns about vaccine safety (higher scores in factor 2). This finding supports the previous literature indicating that reduced regret and increased satisfaction with health decisions are linked to fewer concerns about subsequent health interventions [40]. Additionally, the number of COVID-19 vaccine boosters significantly predicted Concerns. Participants who had received one or more boosters exhibited lower concerns than those who had not, underscoring the consistency of the positive reinforcement effect of repeated vaccination.

Furthermore, being an MScN student versus being a worker as an RN was a significant predictor of Compliance, with MScN students demonstrating higher compliance compared to RNs. This outcome is consistent with studies suggesting that advanced education levels help enhance compliance with health guidelines due to better understanding and greater acceptance of medical recommendations [41,42]. However, it contrasts with other research that has not always found a clear link between educational attainment and compliance behavior [43]. This discrepancy highlights the importance of context-specific factors in shaping compliance behaviors, such as the nature of the healthcare environment and the specific educational curricula.

Previous research has highlighted significant levels of vaccine hesitancy among healthcare professionals in Italy [22,23,24]. In this context, our study introduced the novel insight that decision regret plays a critical role in shaping vaccine hesitancy among healthcare students and workers. While prior studies have focused on general attitudes and beliefs [44], our research quantitatively demonstrated that lower decision regret is associated with higher trust and lower concerns about vaccination. This highlights the importance of addressing emotional and psychological factors in public health interventions, which has been relatively underexplored in the existing literature, acknowledging the complexity of the decision-making process [45].

The findings of this study have several practical implications for public health interventions aimed at increasing vaccination rates among healthcare students and workers. Understanding the predictors of vaccine hesitancy, such as decision regret and the number of vaccine boosters, helps to develop targeted strategies to address these issues. One practical intervention is to focus on reducing decision regret by providing comprehensive and transparent information about the safety and efficacy of vaccines. This could involve educational workshops, seminars, and informational campaigns designed for nursing students and healthcare workers. Addressing the concerns and misconceptions could help build vaccine trust [5]. Additionally, tailored communication strategies that address different groups’ unique needs and perspectives, such as MScN students and RNs, could enhance compliance. For instance, incorporating vaccine education into the curricula of nursing programs and providing continuing education opportunities for RNs could help to ensure that these professionals are well informed and confident in their vaccination decisions.

In addition to the focus of this study on healthcare workers, it is crucial to consider the broader impact of vaccine hesitancy on patients with chronic conditions, where a good example could be represented by patients with Inflammatory Bowel Disease (IBD) [46]. For instance, these patients, like other chronic patients, have been particularly vulnerable during the COVID-19 pandemic, not only due to their heightened risk of severe outcomes but also because of the significant hesitancy surrounding vaccination [46]. This hesitancy often stems from fears that the vaccine may exacerbate their condition or interact negatively with ongoing treatments. Recent studies show that such concerns have led to lower vaccination rates and increased non-adherence to recommended therapies in this population [46]. Addressing vaccine hesitancy among patients with chronic conditions requires targeted interventions that acknowledge and mitigate these specific fears. Healthcare providers should offer tailored counseling that reassures patients about the safety of vaccines in the context of their chronic disease management, emphasizing the importance of vaccination in preventing severe COVID-19 outcomes. Such efforts could include collaboration between clinical specialists and primary care providers to deliver consistent and supportive messaging, potentially integrating telemedicine for ongoing patient education and reassurance. Public health interventions could promote vaccination uptake and adherence across vulnerable populations by extending vaccine education and support to patients with chronic conditions.

Building upon our study’s findings, it is important to contextualize our results within the broader body of the literature on vaccine hesitancy among healthcare students, particularly those in nursing programs [14,47,48,49,50,51,52]. Extensive research has documented that nursing students often exhibit vaccine hesitancy, primarily due to concerns regarding vaccine safety, misinformation, and perceived low risk of contracting vaccine-preventable diseases [24,47,52]. Misinformation and inadequate understanding of vaccines significantly contribute to hesitancy among healthcare students, including nursing students [52,53,54,55,56]. In this regard, targeted educational interventions could significantly diminish vaccine hesitancy among nursing students, underscoring the necessity of incorporating comprehensive vaccine education into nursing curricula [49]. Nursing students’ vaccine hesitancy was previously described as influenced by concerns about side effects, suggesting the importance of addressing these specific concerns through targeted educational programs [52,53,54,55,56]. Furthermore, a strong association between trust in healthcare systems and vaccine acceptance was previously demonstrated [52], which is particularly relevant for nursing students at the frontline of patient care and public health promotion. In light of this evidence from the literature, our research suggests that reducing decision regret through clear communication, supportive interventions, and educational initiatives could be a key strategy in enhancing vaccine uptake among nursing students and healthcare professionals. This highlights the necessity for a multi-faceted approach in public health strategies that addresses both cognitive understanding and emotional well-being, aiming to effectively combat vaccine hesitancy in this critical population.

This study highlights several areas where future research could provide valuable insights into vaccine hesitancy and decision regret, particularly among healthcare students and workers. First, longitudinal studies are needed to assess changes in vaccine hesitancy and decision regret over time. Such studies could track participants across different stages of their education and professional careers, providing a dynamic understanding of how attitudes and behaviors evolve. Longitudinal data would help identify critical periods when interventions might be most effective and determine the long-term impact of initial vaccine experiences on subsequent vaccination decisions. Second, further investigation is required into the specific factors influencing decision regret. While this study identified significant predictors such as the number of vaccine boosters and professional profile, other potential determinants, including cultural, psychological, and social factors, warrant deeper exploration. Understanding these nuances could help develop more personalized interventions. Research should also explore how these factors interact and compound to influence decision regret, which can inform the design of comprehensive strategies to mitigate regret and improve vaccine acceptance. Third, there is a need to explore the impact of targeted interventions on vaccine hesitancy and decision regret in different demographic groups. Future studies should evaluate the effectiveness of various educational and informational interventions tailored to specific groups, such as BScN students, MScN students, and RNs. Comparative studies across different regions and healthcare settings can reveal context-specific challenges and successful strategies, offering insights that can be generalized or adapted to other settings. Finally, it is essential to investigate the role of emotional and psychological support in addressing vaccine hesitancy and decision regret. Future research should examine interventions that include counseling, peer support, and mental health resources to help healthcare workers and students process their vaccination experiences positively. Such approaches may be particularly effective in reducing decision regret and increasing compliance with vaccination recommendations.

This study also has several limitations that should be acknowledged. One significant limitation is the potential for self-report bias. Participants may have provided socially desirable responses or inaccurately recalled their experiences and attitudes, which could affect the validity of the findings. Although measures such as validation checks and anonymity were employed to mitigate this bias, it cannot be entirely eliminated. Another limitation is the cross-sectional design of this study. This design captures data at a single point, limiting the ability to conclude causality and changes over time. Longitudinal studies are needed to understand how vaccine hesitancy and decision regret evolve over time. The generalizability of the findings to other populations is also a concern. This study focused on BScN and MScN students and workers as RNs in Italy, and the results may not be directly applicable to other countries or different healthcare settings. Cultural, socio-economic, and healthcare system differences could influence vaccine hesitancy and decision regret, so caution should be taken when extrapolating these findings to other contexts.

## 5. Conclusions

This study provides a comprehensive analysis of vaccine hesitancy and decision regret among BScN and MScN students and workers as RNs in Italy, highlighting critical factors that influence trust, concerns, and compliance regarding COVID-19 vaccination. The findings reveal the significant role of decision regret and the number of vaccine boosters in shaping attitudes toward vaccination. Lower levels of decision regret were associated with higher trust in vaccine efficacy and benefits and lower concerns about vaccine safety. The number of COVID-19 vaccine boosters emerged as a key predictor, with more boosters correlating with reduced concerns and increased trust, demonstrating the importance of positive reinforcement through vaccination experiences. This study also revealed that MScN students exhibited higher compliance compared to RNs, suggesting that advanced education levels may enhance adherence to vaccination guidelines. These insights emphasize the need for targeted educational interventions to address vaccine hesitancy and decision regret, particularly among different professional profiles within the healthcare sector. Future research is needed to further comprehend the role of decision regret on vaccine hesitancy over time.

## Figures and Tables

**Figure 1 vaccines-12-01054-f001:**
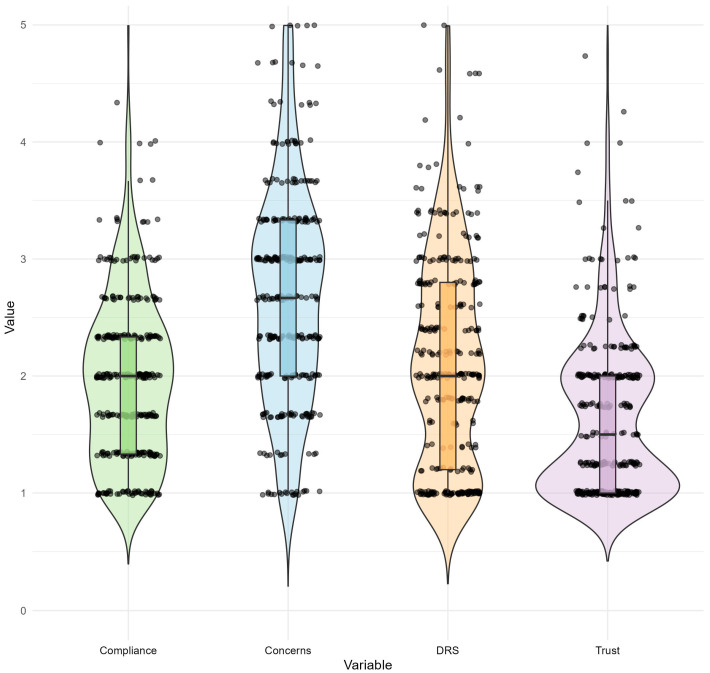
Distribution of DRS and aVHS scores.

**Figure 2 vaccines-12-01054-f002:**
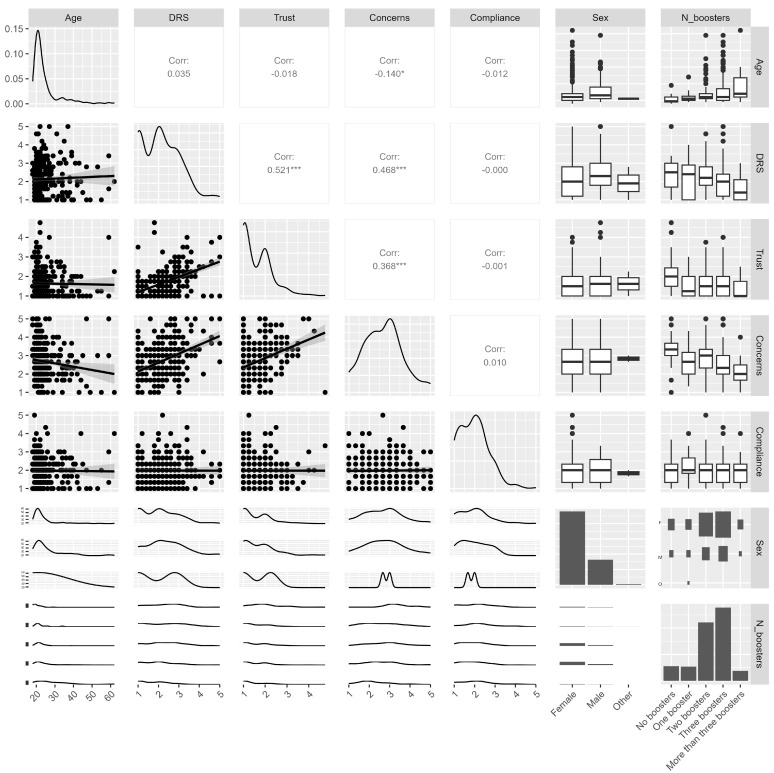
Correlogram. The plot illustrates the relationships between Age, DRS, Trust, Concerns, Compliance, Sex, and the number of boosters. The diagonal plots show the distribution of each variable, with density plots for continuous variables and bar plots for categorical variables. The lower triangle displays scatter plots and smooth density plots, highlighting pairwise relationships, such as the positive correlation between lower decision regret and higher trust. The upper triangle presents Pearson correlation coefficients, summarizing the strength and direction of these relationships, with statistical significance indicated by asterisks. Box plots for categorical variables, such as Sex and Number of Boosters, depict how continuous variables vary across different categories. * indicates *p*-values lower than 0.05, *** indicates *p*-values lower than 0.001.

**Table 1 vaccines-12-01054-t001:** Sample characteristics (N = 324).

	N	%
Sex		
Females	240	74.1
Males	82	25.3
Other	2	0.6
Age		
years (median; IQR)	22	20–25
Marital Status		
Unmarried/living alone	296	91.4
Married	28	8.64
Location		
Northern Italy	169	52.2
Central Italy	68	21.0
Southern Italy	87	26.9
Profile		
BScN student	191	59.0
MScN student	72	22.2
Worker as RN	61	18.8
Work settings (n = 61 workers)		
Emergency or Critical care	7	11.5
Medical care	13	21.3
Surgical care	14	23.0
Primary care	14	23.0
Outpatient services	13	21.3
COVID-19 vaccine *		
Yes	296	91.4
Number of COVID-19 Vaccine Boosters		
No boosters (unvaccinated respondents)	28	8.64
One booster	27	8.33
Two boosters	111	34.3
Three boosters	139	42.9
More than three boosters	19	5.89
Decision Regret Scale (DRS)		
Score (median; IQR)	2	1.2–2.8
Adult Vaccine Hesitancy Scale (aVHS)		
Score of Trust in Vaccine Efficacy and Benefits (median; IQR)	1.5	1.0–2.0
Score of Concerns about Vaccine Safety (median; IQR)	2.67	2.0–3.33
Score of Trust in Health Authorities and Compliance (median; IQR)	2.0	1.3–2.3

Legend: BScN student: Bachelor of Science in Nursing student; MScN student: Master of Science in Nursing student; Worker as RN: Registered Nurse currently working; IQR: Interquartile Range. * Note: Among the 28 individuals who were not vaccinated (reasons not specified), 24 were BScN students (reasons not specified), 3 were MScN students, and 1 was a worker (citing health reasons).

**Table 2 vaccines-12-01054-t002:** Predictors for vaccine hesitancy (Trust, Concerns, Compliance) and Decision Regret Score.

Direct Path	Estimate	SE	*p*-Value
**Trust by (R^2^ = 31.3%)**			
Decision Regret Score	**0.396**	**0.044**	**<0.001**
Age	0.005	0.008	0.564
Location: North Italy (Ref: South Italy)	0.072	0.074	0.327
Location: Central Italy (Ref: South Italy)	0.047	0.095	0.617
Profile: BSN Students (Ref: Workers)	0.160	0.116	0.166
Profile: MSN Students (Ref: Workers)	0.078	0.095	0.408
One Booster (Ref: No boosters)	**−0.531**	**0.205**	**0.009**
Two Boosters (Ref: No boosters)	**−0.475**	**0.197**	**0.016**
Three Boosters (Ref: No boosters)	**−0.397**	**0.197**	**0.045**
More Than Three Boosters (Ref: No boosters)	**−0.483**	**0.222**	**0.030**
**Concerns by (R^2^ = 26.9%)**			
Decision Regret Score	**0.463**	**0.054**	**<0.001**
Age	−0.008	0.013	0.515
Location: North Italy (Ref: South Italy)	−0.153	0.107	0.156
Location: Central Italy (Ref: South Italy)	−0.188	0.134	0.161
Profile: BSN Students (Ref: Workers)	0.165	0.198	0.405
Profile: MSN Students (Ref: Workers)	0.133	0.155	0.393
One Booster (Ref: No boosters)	**−0.656**	**0.227**	**0.004**
Two Boosters (Ref: No boosters)	**−0.458**	**0.189**	**0.015**
Three Boosters (Ref: No boosters)	**−0.575**	**0.187**	**0.002**
More Than Three Boosters (Ref: No boosters)	**−0.715**	**0.260**	**0.006**
**Compliance by (R^2^ = 2.9%)**			
Decision Regret Score	−0.004	0.040	0.916
Age	−0.003	0.012	0.818
Location: North Italy (Ref: South Italy)	0.073	0.091	0.425
Location: Central Italy (Ref: South Italy)	0.154	0.116	0.183
Profile: BSN Students (Ref: Workers)	−0.065	0.190	0.730
Profile: MSN Students (Ref: Workers)	**−0.296**	**0.155**	**0.050**
Female	−0.019	0.089	0.833
**Decision Regret Score by (R^2^ = 4.4%)**			
More Than Three Boosters (Ref: No boosters)	**−0.527**	**0.181**	**0.004**
Age	−0.014	0.010	0.156
Location: North Italy (Ref: South Italy)	−0.093	0.120	0.436
Location: Central Italy (Ref: South Italy)	−0.036	0.136	0.791
Profile: BSN Students (Ref: Workers)	**−0.536**	**0.176**	**0.002**
Profile: MSN Students (Ref: Workers)	**−0.366**	**0.170**	**0.031**

Note: Indirect effects of Decision Regret Score on Trust, Concerns, and Compliance were tested but were insignificant. The model well explained sample statistics: χ^2^_(10, N=324)_ = 16.373, *p* = 0.089; RMSEA = 0.044, 90%CI (0.000–0.080), *p* = 0.554; CFI = 0.969; TLI = 0.855; SRMR = 0.030. Legend: SE = standard error. Bold values are those with *p* < α.

## Data Availability

The data presented in the study are available at the request of the corresponding author due to privacy restrictions. Authors are working on an institutional agreement to allow a repository of the raw data on Zenodo as soon as they receive authorization.

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
