# Peer review of "Decision Regret and Vaccine Hesitancy among Nursing Students and Registered Nurses in Italy: Insights from Structural Equation Modeling"

_vaccines, 2024, doi:10.3390/vaccines12091054_

Round 1
Reviewer 1 Report
Comments and Suggestions for Authors
This study offers a timely analysis of decision regret regarding COVID-19 vaccines in nurses and nursing students in Italy. The researchers employed an online survey with two validated scales (Decision Regret Scale and Adult Vaccine Hesitancy Scale, both translated to Italian) to measure decision regret, trust, concern, compliance, and general demographic information from nursing students (BScNs and MScNs) and nurses (RNs) from across the country. They achieved an impressive response rate of 71% and found interesting correlations and predictions among the variables they analyzed. I found the article to be very well written, well researched, and well organized.
I list some feedback and request minor edits.
- The most significant area of improvement in my opinion is Figure 2. Firstly, the bottom left graph lacks numbers on the y-axis (since it refers to the number of boosters, I believe the numbers should be 0, 1, 2, 3, and 4). I am confused by the graphs that have the same metric on the both the x-axis and the y-axis. For example, what is the bar graph with DRS on the x-axis and DRS on the y-axis showing? It is unclear why this is a bar graph. Likewise, the graphs with age on the both axes, trust on both axes, concerns on both axes, and compliance on both axes are also confusing. Perhaps these issues can be remedied with the addition of a detailed figure legend to explain each graph (the simple figure title on line 328 is insufficient). And perhaps this figure needs to be redesigned or reformatted. I request that the authors consider ways to improve the readability of Figure 2.
- It would be helpful to redefine important acronyms in the first few paragraphs of the Results section, such as Decision Regret Scale (DRS) and adult Vaccine Hesitancy Scale (aVHS). I found myself frequently flipping back to the Methods section to refresh my memory of these scales and their meaning. So a couple reminders in the Results section would be beneficial.
- The second sentence of the first paragraph of the Discussion (lines 356-357) can be edited to improve clarity. See the suggested edit in red below.
"The median DRS score of 2.0 on a 5.0 scale indicated moderate to high levels of regret regarding vaccination decisions."
- Line 359 is missing the 1 RN.
- Overall, there is one explanation related to the number of boosters and the related predictions that I think is missing from the Discussion. If someone received one vaccine and suffered from some side effects or perhaps got severely sick with COVID-19 down the road, they may be less likely to receive a booster, and also perhaps lose trust and feel concerned. Thus, this is one explanation for why the number of boosters is related to trust and concerns. This was slightly explained in lines 372 and 380 and 462, but could be explored further in the Discussion.
Author Response
Comment 1: This study offers a timely analysis of decision regret regarding COVID-19 vaccines in nurses and nursing students in Italy. The researchers employed an online survey with two validated scales (The decision Regret Scale and Adult Vaccine Hesitancy Scale, both translated into Italian) to measure decision regret, trust, concern, compliance, and general demographic information from nursing students (BScNs and MScNs) and nurses (RNs) from across the country. They achieved an impressive response rate of 71% and found interesting correlations and predictions among the variables they analyzed. I found the article to be very well written, well researched, and well organized. I list some feedback and request minor edits.
Response 1: Thank you for appreciating our work and indicating such relevant comments to improve the manuscript. We have tried to address as best your comments in the current version of the manuscript.
Comment 2: The most significant area of improvement in my opinion is Figure 2. Firstly, the bottom left graph lacks numbers on the y-axis (since it refers to the number of boosters, I believe the numbers should be 0, 1, 2, 3, and 4). I am confused by the graphs that have the same metric on the both the x-axis and the y-axis. For example, what is the bar graph with DRS on the x-axis and DRS on the y-axis showing? It is unclear why this is a bar graph. Likewise, the graphs with age on the both axes, trust on both axes, concerns on both axes, and compliance on both axes are also confusing. Perhaps these issues can be remedied with the addition of a detailed figure legend to explain each graph (the simple figure title on line 328 is insufficient). And perhaps this figure needs to be redesigned or reformatted. I request that the authors consider ways to improve the readability of Figure 2.
Response 2: We appreciate your insightful feedback regarding Figure 2. In response, we have made several key revisions to enhance clarity and readability. The y-axis labels in the bottom left graph have been corrected to clearly indicate the number of boosters (0, 1, 2, 3, 4). We have also reformatted the correlogram to replace redundant bar graphs with density plots, providing a clearer representation of the distribution of individual variables. Additionally, we have added a concise and informative explanation next to Figure 2, guiding readers on how to interpret the diagonal plots, scatter plots, correlation coefficients, and the relationships involving categorical variables such as Sex and Number of Boosters. These revisions have improved the visual presentation and made the correlogram more intuitive, addressing the concerns raised. We believe these changes significantly enhance the reader’s understanding of the relationships between the study variables. Thank you for your constructive suggestions.
Comment 3: It would be helpful to redefine important acronyms in the first few paragraphs of the Results section, such as Decision Regret Scale (DRS) and adult Vaccine Hesitancy Scale (aVHS). I found myself frequently flipping back to the Methods section to refresh my memory of these scales and their meaning. So a couple reminders in the Results section would be beneficial.
Response 3: Thank you for this comment. In Table 1, we added the main acronyms to facilitate the reading of the results.
Comment 4:
The second sentence of the first paragraph of the Discussion (lines 356-357) can be edited to improve clarity. See the suggested edit in red below.
"The median DRS score of 2.0 on a 5.0 scale indicated moderate to high levels of regret regarding vaccination decisions."
Response 4: Thank you for this helpful consideration; we edited it as per your suggestion.
Comment 5: Line 359 is missing the 1 RN.
Response 5: Thank you for paying attention to this detail. We amended it as per your comment.
Comment 6: Overall, there is one explanation related to the number of boosters and the related predictions that I think is missing from the Discussion. If someone received one vaccine and suffered from some side effects or perhaps got severely sick with COVID-19 down the road, they may be less likely to receive a booster, and also perhaps lose trust and feel concerned. Thus, this is one explanation for why the number of boosters is related to trust and concerns. This was slightly explained in lines 372 and 380 and 462, but could be explored further in the Discussion
Response 6: Thank you for this valuable comment. We added a discussion on these mentioned aspects.
Reviewer 2 Report
Comments and Suggestions for Authors
I find this work particularly interesting as it examines a relevant theme in this pandemic era. Vaccine hesitancy among healthcare workers indirectly influences that of patients. As this analysis demonstrates, safety concerns are widespread and affect healthcare professionals, such as nurses, who should not possess blind scepticism towards this issue, despite the robust available evidence on vaccine safety.
The methodology is solid, and the presentation of the results is fluent.
I have only one minor comment to improve the manuscript.
- I would add that vaccine scepticism has affected not only healthcare workers but also patients, particularly those with chronic conditions and specific therapies. I would cite a paradigmatic example of patients with IBD, where significant harm from COVID-19 has been observed, and where vaccine hesitancy has been particularly pronounced due to fears of changes in their disease state (https://pubmed.ncbi.nlm.nih.gov/35973931/).
Author Response
Comment 1:
I find this work particularly interesting as it examines a relevant theme in this pandemic era. Vaccine hesitancy among healthcare workers indirectly influences that of patients. As this analysis demonstrates, safety concerns are widespread and affect healthcare professionals, such as nurses, who should not possess blind scepticism towards this issue, despite the robust available evidence on vaccine safety. The methodology is solid, and the presentation of the results is fluent.
Response 1: thank you for this comment.
Comment 2:
I have only one minor comment to improve the manuscript.
I would add that vaccine scepticism has affected not only healthcare workers but also patients, particularly those with chronic conditions and specific therapies. I would cite a paradigmatic example of patients with IBD, where significant harm from COVID-19 has been observed, and where vaccine hesitancy has been particularly pronounced due to fears of changes in their disease state (https://pubmed.ncbi.nlm.nih.gov/35973931/).
Response 2: Thank you for your suggestion. We acknowledge the broader impact of vaccine skepticism beyond healthcare workers, particularly among patients with chronic conditions such as Inflammatory Bowel Disease (IBD). We have incorporated a discussion of this issue, citing the example of IBD patients who have experienced significant harm from COVID-19 and have shown pronounced vaccine hesitancy due to fears of exacerbating their disease state. The reference to the study you mentioned (https://pubmed.ncbi.nlm.nih.gov/35973931/) has been included to highlight the specific challenges faced by this vulnerable population during the pandemic. This addition strengthens the manuscript by emphasizing the pervasive nature of vaccine hesitancy and its severe implications for patient health, especially among those with chronic conditions.
Reviewer 3 Report
Comments and Suggestions for Authors
This study focused on vaccine hesitancy and decision regret about the COVID-19 vaccine among nursing students (BScN and MScN) and Registered Nurses (RNs) in Italy. The aim of the study was to describe decision regret and vaccine hesitancy among these groups and to understand the factors associated with vaccine hesitancy.
The paper is well written, however, there are some clarifications needed:
1. Introduction: It is necessary to enrich the section on the reasons for vaccine hesitancy, specifically among the student population. See, for example:
https://doi.org/10.3390/ejihpe14010003https://doi.org/10.3390/vaccines9060665
https://doi.org/10.1155/2016/4248071
https://doi.org/10.3205/zma001511
https://doi.org/10.1007/s10654-020-00634-3
https://doi.org/10.3390/vaccines8010052
2. Methods: It is necessary to explain in detail the research procedure—how the link to the questionnaire was distributed, in how many universities, and who contacted the universities to obtain permission.
3. Discussion: The discussion is very sparse, with minimal comparison to previous studies despite the numerous studies that have examined vaccine hesitancy and to which the findings of the current research could be compared. The discussion should be enriched with additional studies and explanations, specifically related to students and to nursing.
Author Response
Comment 1:
This study focused on vaccine hesitancy and decision regret about the COVID-19 vaccine among nursing students (BScN and MScN) and Registered Nurses (RNs) in Italy. The aim of the study was to describe decision regret and vaccine hesitancy among these groups and to understand the factors associated with vaccine hesitancy.
The paper is well written, however, there are some clarifications needed:
- Introduction: It is necessary to enrich the section on the reasons for vaccine hesitancy, specifically among the student population. See, for example:
https://doi.org/10.3390/ejihpe14010003
https://doi.org/10.3390/vaccines9060665
https://doi.org/10.1155/2016/4248071
https://doi.org/10.3205/zma001511
https://doi.org/10.1007/s10654-020-00634-3
https://doi.org/10.3390/vaccines8010052
Response 1: thank you for this comment. We have enriched the introduction by acknowledging the contribution provided by the suggested articles.
Comment 2: Methods: It is necessary to explain in detail the research procedure—how the link to the questionnaire was distributed, in how many universities, and who contacted the universities to obtain permission.
Response 2: thank you for this request. This revision adds specific details about the number of universities and healthcare organizations involved in each geographic area, providing a clearer understanding of the study's methodology, the efforts made to ensure a representative sample, and the other key aspects highlighted in your comment.
Comment 3: Discussion: The discussion is very sparse, with minimal comparison to previous studies despite the numerous studies that have examined vaccine hesitancy and to which the findings of the current research could be compared. The discussion should be enriched with additional studies and explanations, specifically related to students and to nursing.
Response 3: Thank you for your feedback. We have revised the discussion section to address the concerns raised. The updated discussion now includes a more comprehensive comparison with previous studies on vaccine hesitancy, particularly those focusing on healthcare students and nursing students. In the revised version, we incorporated findings from key studies that explored factors contributing to vaccine hesitancy, such as safety concerns, misinformation, and perceived low risk among nursing students. The discussion has been enriched with these additional studies and explanations, providing a more thorough comparison with the existing literature. This enhancement demonstrates how our findings align with and contribute to the broader understanding of vaccine hesitancy in nursing students, highlighting the need for educational strategies that integrate both cognitive understanding and emotional well-being to improve vaccine uptake. Thank you for your guidance, and we believe the revised discussion now offers a stronger, more well-rounded analysis of the findings.
Reviewer 4 Report
Comments and Suggestions for Authors
This study is well designed, properly executed, but too difficult to read and comprehend. It is too long, too detailed and the results (e.g. figure 2) are difficult to understand. The detailed description of the analysis is beyond the comprehension of the average reader.
Comments on the Quality of English LanguageEditing necessary
Author Response
Comment 1: This study is well designed, properly executed, but too difficult to read and comprehend. It is too long, too detailed and the results (e.g. figure 2) are difficult to understand. The detailed description of the analysis is beyond the comprehension of the average reader.
Response 1: Thank you for your feedback and for acknowledging the design and execution of our study. We understand your concerns regarding the readability and complexity of the manuscript, particularly the challenges in interpreting Figure 2 and the detailed analysis. In response to similar feedback from Reviewer 1, we have added a detailed explanation of how to interpret Figure 2, making it more accessible to readers. This explanation is intended to guide the reader through the figure and help them understand the relationships and data presented. Moreover, we have made a concerted effort to streamline the text throughout the manuscript. While we aimed to retain all necessary elements to ensure transparency and comprehensiveness, we have revised sections to improve clarity and readability. This includes simplifying the language where possible and ensuring that the essential information is conveyed effectively without overwhelming the reader. We believe these adjustments will make the manuscript more accessible while maintaining the rigor and transparency that are essential for scientific communication. We appreciate your constructive comments and hope that these revisions address your concerns.
Round 2
Reviewer 4 Report
Comments and Suggestions for Authors
In spite of the effort to make the manuscript more readable, I still think this is too much for the average reader of this journal.
Comments on the Quality of English LanguageSome proof reading is needed.
Author Response
Comment 1:
In spite of the effort to make the manuscript more readable, I still think this is too much for the average reader of this journal.
Response 1:
Thank you for your feedback on the readability of our manuscript. We appreciate your concern regarding the content’s accessibility to the journal’s broader audience. In response to your comments, we have undertaken a thorough revision to enhance the clarity and readability of the manuscript. This revision included comprehensive grammar and English proofreading to ensure the text is clear and accessible.
Despite these efforts to enhance readability, we believe it is crucial to retain certain technical details that are foundational to the transparency and integrity of our study, particularly regarding the Structural Equation Modeling (SEM) and validation tests employed. SEM is a powerful tool in our analysis, and accurate reporting is essential to allow replication and to convey the robustness of our findings. We have included comprehensive details on SEM, such as the goodness-of-fit indices (e.g., RMSEA, CFI, TLI), standard errors, and information on estimates because these are essential for evaluating the validity and reliability of our model. In the validation of the Adult Vaccine Hesitancy Scale (aVHS), we performed an Exploratory Factor Analysis (EFA) and reported the factor loadings, internal consistency (McDonald’s ω), and dimensionality of the scale. While this information is technical, it is presented in a way that supports the study’s conclusions and is essential for the transparency of our methodology.
We understand the importance of making our research accessible to a broad readership, including clinicians who may not be specialists in SEM but are interested in the study’s implications for practice. To this end, we have ensured that our results are presented in a clear and understandable manner. For example, while we include detailed SEM outputs in the results section, we also provide a narrative explanation that interprets these results in layman’s terms, helping readers understand the practical significance of our findings.
In discussing the influence of decision regret on vaccine hesitancy, we have included a simplified explanation in non-technical terms: For example, “[…]lower decision regret is associated with higher trust in vaccine efficacy and fewer concerns about vaccine safety. This means that individuals who are more confident in their decision to get vaccinated are less likely to have doubts about the vaccine’s safety […]” is understandable for everybody. This explanation is designed to be easily understood by readers without a statistical background, while the accompanying technical details ensure transparency.
We believe this version strikes an appropriate balance, ensuring that the manuscript is both accessible to a broader audience and informative for readers with a more technical interest in SEM. In addition, we tested the readability of the manuscript by asking two clinical colleagues who do not have high expertise in statistical analysis if they understand its meaning. Both answered that they understood what we presented in terms of methods, results, and discussion.
We hope this revised manuscript meets the expectations of the journal’s readership and that the technical rigor we have maintained will be appreciated for its contribution to the field.